# Harnessing Omics Approaches on Advanced Preclinical Models to Discovery Novel Therapeutic Targets for the Treatment of Metastatic Colorectal Cancer

**DOI:** 10.3390/cancers12071830

**Published:** 2020-07-08

**Authors:** Manuela Porru, Pasquale Zizza, Nadia Panera, Anna Alisi, Annamaria Biroccio, Carlo Leonetti

**Affiliations:** 1UOSD SAFU, Department of Research, Advanced Diagnostic, and Technological Innovation, IRCSS-Regina Elena National Cancer Institute, 00144 Rome, Italy; manuela.porru@ifo.gov.it; 2UOSD Oncogenomic and Epigenetic Unit, Department of Research, Advanced Diagnostic, and Technological Innovation, IRCSS-Regina Elena National Cancer Institute, 00144 Rome, Italy; pasquale.zizza@ifo.gov.it; 3Research Unit of Molecular Genetics of Complex Phenotypes, Bambino Gesù Children’s Hospital, IRCCS, 00165 Rome, Italy; nadia.panera@opbg.net (N.P.); anna.alisi@opbg.net (A.A.)

**Keywords:** colon cancer, anti-EGF, new therapies, organoids, PDXs, orthotopic tumors, GEMM, omics technologies

## Abstract

Metastatic colorectal cancer (mCRC) remains challenging because of the emergence of resistance mechanisms to anti-epidermal growth factor receptor (EGFR) therapeutics, so more effective strategies to improve the patients’ outcome are needed. During the last decade, the application of a multi-omics approach has contributed to a deeper understanding of the complex molecular landscape of human CRC, identifying a plethora of drug targets for precision medicine. Target validation relies on the use of experimental models that would retain the molecular and clinical features of human colorectal cancer, thus mirroring the clinical characteristics of patients. In particular, organoids and patient-derived-xenografts (PDXs), as well as genetically engineered mouse models (GEMMs) and patient-derived orthotopic xenografts (PDOXs), should be considered for translational purposes. Overall, omics and advanced mouse models of cancer represent a portfolio of sophisticated biological tools that, if optimized for use in concert with accurate data analysis, could accelerate the anticancer discovery process and provide new weapons against cancer. In this review, we highlight success reached following the integration of omics and experimental models; moreover, results produced by our group in the field of mCRC are also presented.

## 1. Introduction

Worldwide, colorectal cancer (CRC) is the third most common cancer with more than 1.8 million cases accounting for about 10% of all annually new cancer diagnoses and the third cause of cancer-related deaths with a mortality rate of 880,000 people [1]. Considerable progress in the prognosis of metastatic CRC (mCRC) was made from the end of the last century when the overall survival was about 12 months to the current 30 months. Several factors have contributed to the increase of the mCRC patients’ outcomes, including the introduction of large-scale screening programs, the surgical resection of liver metastases, and the optimization of systemic chemotherapy (FOLFOX, FOLFIRI, or FOLFOXIRI). Moreover, the introduction of biological therapies based on the use of monoclonal antibodies (mAbs) against either epidermal growth factor receptor (EGFR)—cetuximab and panitumumab—or vascular endothelial growth factor-A (VEGF-A)—bevacizumab—significantly improved patients’ survival [2].

Notwithstanding this advancement, the treatment of mCRC represents a great challenge for oncologists, as the five-year survival rate is less than 15% [1]; thus, the identification of more effective therapeutic strategies is of paramount importance. From this point of view, the use of complex omics technologies is particularly relevant to deeply investigate the molecular landscape of mCRC. Moreover, the integration of omics data with the functional validation of druggable targets in experimental preclinical models able to recapitulate heterogeneity of mCRC could be an excellent approach for the discovery and development of novel therapeutics.

In this review, we give an overview the state of art and discuss recent progress in this field of research. Moreover, to make the design and analysis of in vivo experiments more powerful for clinical application, the application of the Response Evaluation Criteria In Solid Tumors (RECIST) of mice is also discussed.

## 2. Validated Biomarkers of Response to Anti-EGFR mAbs Treatment

A key piece of data in the treatment of mCRC was represented by the approval in 2004 of cetuximab, a monoclonal antibody specific for the extracellular ligand-binding domain of EGFR. This receptor is over-expressed in 60–80% of tumors and has a pivotal role in the growth and progression of mCRC, and, when it is over-expressed, is associated with poor prognosis [3]. Randomized clinical trials, that followed promising pre-clinical studies, demonstrated that cetuximab was effective in combination with irinotecan in over-expressing EGFR mCRC patients who previously did not respond to single therapy with irinotecan [4]. Importantly, subsequent studies showed that the efficacy of cetuximab and panitumumab (approved in 2006) was limited to the patients with *RAS* wild-type (WT) tumors, as mutations in EGFR signaling may induce receptor-independent pathways that render the tumors unresponsive to anti-EGFR-based therapy. In fact, activating mutations in hot spot regions of exons 2, 3, and 4 of the *KRAS* or of *NRAS* genes, reported in more than 50% of mCRC cases, were responsible for the primary resistance to anti-EGFR therapy and represented the major exclusion criteria for treatment of patients with these drugs [5].

Only 30–40% of mCRC patients unresponsive to anti-EGFR treatment have been found to be positive for *RAS* mutations [6]. In addition, it was reported that EGFR signaling activation affected not only the RAS/RAF/MAPK+ pathway but also the PTEN-PI3K-Akt cascade [7]. All of these observations have prompted investigators to analyze the involvement in mCRC resistance to anti-EGFR mAbs of other genes of the RAS/RAF/MAPK and PI3K/PTEN/Akt pathways. In particular, a mutation in the *BRAF* gene, which encodes a downstream effector of KRAS, resulting in the constitutive activation of the MAPK pathway, was found in about 6% of mCRC patients with no overlap with the *RAS* mutation [8]. In particular, a retrospective study by Di Colantonio [9], performed in 113 tumor samples from mCRC patients treated with cetuximab or panitumumab, demonstrated that the presence of the *BRAF* V600E mutation impaired the response to anti-EGFR treatment. The role of this mutation in interfering with the efficacy of anti-EGFR was also confirmed in tumor cells naturally or experimentally bearing the BRAFV600E mutation. Further retrospective and meta-analyses studies have more recently confirmed the prognostic value of *BRAF* as a biomarker for predicting the efficacy of anti-EGFR mAbs therapy [2,10].

On this basis, according to international guilines, the analysis of alterations in the *RAS* and *BRAF* genes is mandatory for selecting patients eligible for anti-EGRF mAbs treatment, thus leading to clinical and economic advantages.

## 3. Emerging Biomarkers of Response to Anti-EGFR mAbs Treatment 

Other than mutational status of *RAS* and *BRAF*, further biomarkers are emerging as possible responsible for the innate resistance to anti-EGFR mAbs, such as alterations in the parallel or downstream pathways of the EGFR signaling.

HER2: The human EGFR-2 (*HER2*), an oncogene that encodes for a transmembrane glycoprotein receptor, functions as an intracellular tyrosine kinase and is involved in the activation of two signal transduction pathways (RAS-RAF-ERK and PI3K-PTEN-AKT). *HER2* gene amplification was reported in 7% of patients with CRC [11], and this was associated with the resistance of the tumor to anti-EGFR therapy [12]. The authors used an approach that was, at that time, very innovative and was based on a cohort of 85 mCRC genetically characterized patient-derived xenografts (PDXs) that recapitulate the molecular heterogeneity observed in clinical studies. In particular, the authors observed that PDXs harboring wild-type *KRAS/NRAS/BRAF/PIK3CA* genes but *HER2* amplification were resistant to cetuximab. However, the administration of trastuzumab (an anti-HER2 mAb) or lapatinib (a tyrosine kinase inhibitor effective against HER2) was able to inhibit tumor growth, demonstrating the role of HER-2 in primary resistance to anti-EGFR mAbs. These observations were subsequently confirmed both in CRC cells transfected with activating *HER2* gene mutations and in mice-bearing *HER2*-mutated xenografts [13]. Later, further clinical studies confirmed the role of *HER2* amplification in predicting response to anti-EGFR treatment. In particular, a retrospective study demonstrated that in *RAS/BRAF* WT mCRC patients without *HER2* amplification, the anti-EFGR treatment produced a significantly longer progression-free survival (PFS) than patients harboring *HER2* amplification [14]. While these observations need to be confirmed in larger studies, the study suggested the role of *HER2* amplification as a negative biomarker of response just like *RAS* mutations and that the assessment of *HER2* amplification, in addition to HER2 mutations, could be critical to therapeutic decisional tree.

c-MET: This protein, also known as tyrosine protein kinase MET or mesenchymal-epithelial transition factor, has a key role in initiating a range of signals that regulate various cellular functions and has been suggested to be involved with the growth, survival, and progression of CRC [15]. *c-MET* overexpression and/or amplification has been associated not only with poor outcomes, especially in patients at advanced stage [16], but also as a predictor of resistance to anti-EGFR-mAbs. To this purpose, Bardelli’s group reported that *KRAS* WT patients, who did not respond to cetuximab or panitumumab, were characterized by *MET* amplification [17]. Based on these observations, they deeply investigated the relationship between MET amplification and resistance to anti-EGFR therapy [12]. The authors confirmed that mCRC PDXs with *KRAS*, *BRAF*, *NRAS*, or *PIK3CA* mutated or with the amplification of *HER2* were resistant to cetuximab treatment. At the same time, the study highlighted that the presence of these alterations did not correspond with the number of PDXs that were resistant to cetuximab. Interestingly, they observed that in the population of PDXs WT for *KRAS*, *BRAF*, *NRAS*, *PIK3CA*, and *HER2,* a significant fraction of the tumors resistant to cetuximab were characterized by *MET* amplification. These observations were confirmed in cell lines and in xenografts WT for *KRAS*, *BRAF*, *NRAS*, *PIK3CA*, and *HER2*, with the ectopic overexpression of *MET*. Importantly, the use of a specific anti-MET inhibitor restored the sensitivity of these cells, thus corroborating the role of *MET* in driving resistance to anti-EGFR mAbs [17].

PTEN: The phosphatase and tensin homologue deleted on chromosome ten (*PTEN*) is a tumor suppressor gene that acts through the negative regulation of the PI3K/Akt pathway, another pathway that is associated to EGFR signaling. *PTEN* loss, by germinal or somatic mutations, has been observed in 30% of CRC [18], but its role in predicting the clinical benefit to anti-EGFR treatment has not been fully ascertained. In fact, Frattini and colleagues [19] reported that when patients had intact PTEN, most of them responded to cetuximab-based chemotherapy, while no benefits were observed in patients with *PTEN* loss, notwithstanding the *EGFR* gene amplification and the presence of *KRAS* WT. In contrast, Karapetis and co-workers [20] reported that there was a no positive correlation between loss of *PTEN* and lack of response to anti-EGFR mAbs. In fact, it was observed in this study that patients with PTEN protein expression was not predictive of benefit from treatment, both in the whole study population and in the *KRAS* WT subset.

PIK3CA: The role of alterations in the p110α subunit of *PI3K* gene (*PIK3CA*) is still under investigation. *PIK3CA* mutations are reported in exons 9 and 20 and occur in 30% of CRC cases [21]; the activation of p110α leads to the production of phosphatidylinositol 3,4,5-triphosphate (PIP3), which, in turn, is a substrate of PTEN [22]. A key contribution in this field of research came from the work by Sartore-Bianchi [23], which identified activating mutations in the lipid kinase *PIK3CA* gene as an independent biomarker for the unresponsiveness of *KRAS* WT mCRC to anti-EGFR mAbs. In fact, the authors found that patients with *PIK3CA* mutations in exons 9 and 20 did not achieve objective tumor response after treatment with panitumumab or cetuximab. In the same period, Prenen and colleagues [24] did not confirm the role of *PIK3CA* mutations in predicting the resistance to anti-EGFR therapies. In fact, in a group of 200 mCRC patients treated with cetuximab, the authors did not find a correlation between *PIK3CA* mutations and resistance to the treatment, since the percentage of patients with *PIK3CA* mutations was almost the same in responders and non-responders. At the same time, in selected *KRAS* WT patients, the response rate of patients with *PIK3CA* mutations was superimposable to that bearing *PIK3CA* WT.

Finally, the role of PTEN/PIK3CA as independent predictive marker of response to anti-EGFR mAbs has been difficult to establish since loss of *PTEN* and mutations in *PIK3CA* are frequently associated with mutations in *KRAS* or *BRAF* [20]. Thus, on the basis of current knowledge, it is possible to affirm that the combination of mutation of *PIK3CA* (namely exon 20) and *PTEN* loss with the *BRAF* mutation predict a worse outcome in a context of *KRAS* WT mCRC patients [25]. At the same time, a very large Italian study demonstrated that patients with *KRAS*, *NRAS*, *BRAF*, and *PIK3CA* WT genes (quadruple WT tumors) have the greatest benefit from chemotherapy plus cetuximab with respect to patients with a mutation in at least one of these genes [26].

In conclusion, this is an expanding research field, and additional mechanisms of mCRC resistance to EGFR blockades are under investigation. Findings from these innovative studies may help to better define the subset of patients who benefit from anti-EGFR therapy.

## 4. Secondary Resistance to Anti-EGFR Treatment 

A major problem in the context of mCRC management is that despite the initial response to the anti-EGFR therapy of *RAS-BRAF* WT patients, the onset of relapse of the disease has been observed in these patients following the acquisition of the so-called secondary or acquired resistance to anti-EGFR mAbs treatment.

Both primary and, particularly, secondary resistance are characterized by an intrinsic genetic intra- and inter-tumor heterogeneity, thus highlighting that the knowledge of the molecular landscape of mCRC is necessary for identifying new and more effective treatments for mCRC [27]. In particular, the discovery of circulating DNA (ctDNA), molecules of DNA released by tumor cells in the blood of cancer patients, paved the way to the development of a novel and more sensitive diagnostic/prognostic tool to detect, in a rapid and non-invasive manner, molecular abnormalities accounting for resistance to anti-EGFR therapy [28]. Importantly, the analysis of ctDNA in the blood of patients is undoubtedly advantageous respect to biopsy procedures, in terms both of less invasiveness and ethical issues, which limit the number of further biopsies. Moreover, it has been reported that the genetic profile of tumors can change in the course of therapy and that the molecular characteristics of primary tumors and metastases are not always concordant and can change during the course of the disease due to intrinsic molecular cancer heterogeneity [29,30]. The importance of the use of liquid biopsy in the identification of mechanisms involved in the development of resistance to anti-EGFR mAbs was demonstrated in two seminal papers in which *KRAS* mutations was detected in the serum of *KRAS* WT patients who developed resistance to cetuximab or panitumumab treatment [31,32].

To the best of our knowledge, the first paper in which an extensive characterization of acquired resistance of mCRC to anti-EGFR mAbs was performed by analyzing ctDNA in the blood of patients was published by Pietrantonio and co-workers [33]. These authors investigated a group of patients who initially responded to cetuximab or panitumumab-based treatment but subsequently elicited a radiologically-documented disease progression. Tissue samples were analyzed with a next-generation sequencing (NGS) panel that includes 50 genes’ hotspot regions and with in situ hybridization and immunohistochemistry. At the same time, liquid biopsy was performed on plasma, and Droplet Digital PCR was used to identify mutant DNA alleles. The authors reported the presence of *RAS* mutations and *HER2/MET* amplification detected at high frequency in both tissue and blood sample analysis. On the other hand, mutations in *BRAF* and *PIK3CA* were rare in tissue and liquid biopsies, thus suggesting that the PIK3CA/AKT/mTOR pathway could be less crucial for the development of acquired resistance to anti-EGFR mAbs treatment. In comparison to the biopsy, in some cases, mutations in ctDNA were detected at very low levels, suggesting that only a small fraction of tumor cells release DNA as a consequence of heterogeneous resistance mechanisms. The authors concluded that a combination of tumor and liquid biopsies could help to better characterize the mechanism of acquired resistance to anti-EGFR mAbs. It is of note that the heterogeneity of intra-lesion and inter-lesion reinforces the concept that the liquid biopsy could represents a promising method to identify genetic alterations that account for the acquired resistance of mCRC [34,35,36]. Anyway, due to several concerns that need to be addressed before liquid biopsies could be translated into clinical application [37,38], conventional biopsies still remain the gold standard for patient evaluation.

## 5. Advanced Preclinical Models and Omics in the Discovery of New Strategies against mCRC

The described progress in the knowledge of resistance mechanisms characterizing mCRC needs the development of novel and more effective therapeutic strategies. In this view, preclinical research plays a fundamental and irreplaceable role. To date, a major limit of translational medicine is that less than 5% of compounds selected in preclinical studies are really effective and sufficiently safe in clinical trials. There could be several reasons for this high drug attrition rates, but there is a consensus that these limits may be bypassed through the introduction of advanced preclinical models [39].

Established cell lines have contributed to the understanding of the biological and molecular bases of cancer, leading to the identification of effective compounds still employed in clinical practice [40]. A drawback of this model is that in the preclinical studies, only a limited number of cell lines can be employed (usually less than 10), thus limiting the possibility to reproduce a genetic diversity that is typical of patients. An intelligent strategy to overcome this limit was proposed by Medico and colleagues [41]. The authors generated a large panel of established cell lines whose molecular profile was able to entirely cover the clinical features of CRC [42]. In particular, they demonstrated that the most common mutations observed in clinical specimens (e.g., *KRAS*, *NRAS*, *BRAF*, *PIK3CA*, and *PTEN*) were present with the same mutational rate in the selected cell lines. Importantly, the cell lines matched the sensitivity or resistance to anti-EGFR mAbs according to mutational status reported in clinical studies. Notably, this cell collection permitted the authors to demonstrate that the targeting of alterations in kinase genes such as *ALK*, *FGFR2*, *NTRK1/2*, and *RET* (commonly observed in patients unresponsive to anti-EGF Abs) would represent a valid therapeutic strategy against mCRC. Very recently, the same group [43] confirmed their findings in a panel of 29 CRC lines (defined by the authors as xeno-cell lines—XL) derived from a cohort of CRC PDXs obtained by implanting fresh specimens from primary tumors or metastases. Moreover, using these experimental models, the authors validated ERBB2 as possible therapeutic targets sensitive to treatment with lapatinib and trastuzumab [43]. Importantly, this drug combination was proven to be effective against metastatic HER2^+^ CRC in a phase II clinical trial [44].

However, even though the selected collection of cell lines resembles the genetic and transcriptional alterations present in human tumors, these cells grow as a monolayer in a two-dimensional (2D) system and fail to mimic in vivo conditions because most of physiologic characteristics of tumors—including biochemical networks, cell-to-cell interactions, and cell-to-matrix interactions—are lost. There are several papers demonstrating that the culture of tumor cells in a three-dimensional (3D) system, otherwise said spheroids or tumoroids, are biologically relevant for preclinical drug development. Moreover, since tumor cells interact with extracellular matrix (ECM) and tumor stroma in 3D system, the signaling pathway could be influenced because it occurs in tumors in situ [45,46].

In a recent work aimed at identifying biomarkers of sensitivity to anti-EGFR treatment [47], Schütte and co-workers proposed an experimental platform called OncoTrack (OT), derived from a large biobank of 106 CRC that includes 35 organoids and 59 PDXs, from which nineteen tumors were modelled in both systems. By using multiple sequencing approaches (i.e., whole genome, exome, and RNA sequencing), they observed that the genetic profiles of the models were generally concordant with their matched donor tumors; in some cases, divergences due to the intra-tumor heterogeneity were observed.

A limit of the PDX model is that it does not mimic the stages of CRC growth and progression, since tumors are implanted in unnatural heterotopic site (usually the subcutis of the back of mice), and it is well-know that metastatic dissemination is not observed. In contrast, the exploitation of the full potential of a new therapy against mCRC should rely on the orthotopical implantation of tumors in the colon, which could better represent the human clinical situation, both in terms of site of the origin of CRC and in the metastatic spreading.

Several authors have successfully demonstrated the feasibility of this strategy, which permits to follow primary tumor’s growth, metastatic dissemination and efficacy of therapeutics by in vivo imaging.

In particular, the Hoffman’s group developed orthotopic models of mCRC via the implantation of human tumor tissues into the serosa of the cecum of immunosuppressed mice. The growth of primary tumors and spontaneous metastases were visualized by the use fluorophore-conjugated anti-CEA antibodies [48] or by transfecting cells with green or red fluorescence protein [49,50]. This technology represents a very potent tool for the identification of the anticancer activity of new therapeutic approaches.

In order to study the role of the most common mutations associated with the metastatic dissemination of CRC and to dissect the adenoma–carcinoma sequence of CRC, the Van Rheenen’s group developed a model of spontaneous metastases in mice through the orthotopic injection in the cecum of mice of organoids engineered to harbor different combinations of CRC mutations [51,52]. They found that this model mimicked the human CRC condition because primary tumors exhibited a long latency period, and tumor cells were capable of invading the blood circulation and forming metastases at distant sites. Moreover, the real time imaging by intravital microscopy and bioluminescence allowed for the visualization of in vivo tumor cell dynamics. In particular, they generated human CRC organoids with a combination of mutations of the four most frequently gene alterations reported in CRC by using CRISPR/Cas9 genome editing, such as *KRAS* G12D mutation and the knockout of *APC*, *P53*, and *SMAD4*, while *WNT*, *EGFR*, *P53*, and *TGF-β* were present in the normal status. By using this technology, they obtained a number of so-called quadruple-mutant organoids in which one of the four genes was in the WT form and the other three were in mutated forms. Their results demonstrated that the co-presence of mutations in the four genes accelerated the growth of tumors, favored cell migration and the formation of metastases at distant sites, including the liver.

Other groups, using organoids derived from genetically engineered mouse models (GEMMs), WT organoids engineered ex vivo, and patient-derived human CRC organoids, have demonstrated that the orthotopic implantation of CRC in animals recapitulates CRC progression in humans and could be used to investigate the potential of new therapeutics for mCRC [53].

The key question about the ability of experimental systems used to predict the response of humans to therapeutics remains to be solved. In this regard, Vlachogiannis and colleagues [54] approached this problem for a large biobank of metastatic gastrointestinal tumors including mCRC and compared the results obtained in ex vivo organoids and patient-derived orthotopic xenografts (PDOXs) with the response to anticancer therapeutics in clinical trials. Their results obtained by NGS, whole genome sequencing and transcriptomic analysis showed a high concordance between parental biopsies and organoids at successive passages of organoids in culture, and they confirmed that organoids maintain the profile of the original human tumors, with a 96% overlap in the mutational spectrum. Moreover, the response of both organoids and of PDOXs to regorafenib matched the sensitivity or resistance observed in patients, thus confirming the predictive value of these models.

The use of organoids and their implantation in animal models is an expanding field of investigation in preclinical research [55], but for the extensive application of this platform in personalized medicine, the implementation of studies that demonstrate the concordance between preclinical response and patient outcomes is necessary.

Overall, it is possible to assert that the currently available pre-clinical models (ranging from established cell lines in vitro to the most advanced in vivo models) are distinguished by a number of characteristics that make them unique and irreplaceable (Figure 1). However, since none of these experimental models are capable of fully recapitulating the complexity of human cancer, the most valuable strategy to improve our knowledges is to combine multiple experimental approaches and then integrate the obtained results.

## 6. Development of Novel Therapeutics in Targeting mCRC: The Experience of Our Laboratory 

In the last decade, several new molecules have been investigated for their antitumoral activity against mCRC that is resistant to standard therapies. In this view, G-quadruplex (G4)-ligands—a class of small molecules able to bind and stabilize non-canonical secondary structures of nucleic acids—has received particular attention. Indeed, a number of studies from our and other laboratories have shown that these molecules exert potent antitumoral activity [56,57,58,59]. Recently, we demonstrated that EMICORON, a G4 ligand synthetized in our laboratory, is effective in the treatment of mCRC [60]. In particular, by using advanced CRC preclinical models (i.e., PDX and orthotopic models), we showed that EMICORON is able to inhibit the growth of primary tumors and to negatively affect the dissemination of tumor cells to lymph nodes, the intestine, the stomach and the liver, presenting a potent antitumoral activity while exhibiting a very favorable toxicological profile [60,61].

In subsequent studies, we found that PDXs originating from the same patient can show different responses to treatment with EMICORON. Indeed, while some mice showed a complete remission, in others, there was no-response or an initial response followed by disease recurrence and rapid progression [62], reflecting the heterogeneity within the tumor of origin.

Starting from these results, we are now pointing at defining the mechanism(s) underlying the responsiveness/resistance of colorectal cancer to EMICORON. Nowadays, large-scale analyses—the so-called omics approaches—represent the most potent tools to obtain a comprehensive understanding of tumor biology and, in particular, of all those mechanisms participating in tumor formation, development, dissemination, and drug resistance. Based on these considerations and knowing that G4-ligands can affect the expression of genes containing G4 structures within (or in proximity of) their promoter, we performed an extensive analysis of gene expression based on the TaqMan OpenArray (Life Technologies, Thermo Fisher Scientific Corporation, Foster City, CA, USA). The analysis, performed on a panel of 624 genes, provided extensive information regarding the transcriptional profile of tumor cells treated with EMICORON. Conscious of the importance of adopting multiple investigation models, experiments were performed on both RNA extracted from PDXs and established CRC cell lines.

The first set of analyses performed on PDXs with a complete response to EMICORON (Figure 2A) showed that EMICORON affected the expression of 163 genes (30 up-regulated and 133 down-regulated). Subsequently, the analyses were also extended to unresponsive PDXs. This second set of data evidenced the up-regulation of 179 genes and the down-regulation of 74 genes (Figure 2B).

In order to identify the genes really modulated by EMICORON, the results of the two experiments were matched, and the genes common to the two analyses were subtracted from the list of genes modulated in the responder PDXs (Figure 3A,B). Based on this criterion, the number of genes putatively affected by EMICORON treatment dropped to 82 (11 up-regulated and 71 down-regulated genes; see Figure 3A,B).

Transcriptional analyses were then extended also to HCT116, a well-established CRC cell line. Interestingly, the results of OpenArray identified 153 genes (67 up-regulated and 86 down regulated) modulated in response to EMICORON treatment (1 µM for 24 h) (Figure 2C). Since these results were reasonably different from those obtained in the PDXs, the genes modulated in the HCT116 cells were compared with those resulting from the first analysis, and the sole common genes were selected (Figure 3C,D). Through these analyses, five up-regulated and 19 down-regulated genes, currently under validation, were finally identified.

To establish an order of priority in targets’ validation, all the genes modulated in response to EMICORON were analyzed by QGRS Mapper (http://bioinformatics.ramapo.edu/QGRS/index.php), a bioinformatic tool used for the identification of nucleotide sequences able to generate potential G4 structures. The analyses, carried on a region of 1000 bp upstream of the transcription starting site, revealed that three out of five up-regulated genes (*ACY1*, *MX1,* and *PRDX4*) and 11 out of 19 down-regulated genes (*BMP6*, *BTK*, *EZR*, *F2R*, *FGF3*, *MYCN*, *NID1*, *SHH*, *TLR2*, *TRADD*, and *WNT5A*) contain at least one putative G4 within their promoter, a structural characteristic that would justify the transcriptional control exerted by EMICORON treatment (Figure 3C,D). Based on the data available in the literature, genes known to play a relevant role in CRC will be selected for future studies. In particular, some genes that seem to be promising for their potential as molecular target in mCRC are detailed below. 

The BTK (Bruton’s tyrosine kinase) is a non-receptor tyrosine kinase that is constitutively expressed in B cells, where it plays a key role in promoting the maturation, differentiation, and proliferation of B lymphocytes [63]. Ibrutinib—a BTK-specific inhibitor—was found to be effective against some types of lymphoma and chronic lymphocytic leukemia, also in combination with chemotherapeutics [64,65]. Recently, the overexpression of BTK has been detected in various solid tumors and has been found to be associated with worse prognoses [66]. In particular, a new BTK isoform, p65BTK, has been found to be abundantly expressed in CRC where, acting downstream of the RAS/MAPK pathway, it plays a role as mediator of RAS-induced transformation [67]. Interestingly, BTK inhibition re-sensitizes drug-resistant colon cancer cell lines, organoids, and xenografts lacking TP53 in 5-fluorouracil [68]. Therefore, these evidences could suggest that BTK could be a promising new target in CRC tumors that are resistant to conventional therapy.

Ezrin is a member of the ezrin–radixin–moesin (ERM) family of proteins, which link the actin-containing cytoskeleton to plasma membrane proteins and activate the actin cytoskeleton [69]. Ezrin plays a role in tumorigenesis, invasion, and metastatic processes in a variety of human cancers including colorectal cancer [70]. In a proteomic analysis, ezrin was described as the most promising candidate predictive marker for predicting lymph node metastasis in CRC [71].

Wnt5a is a member of the non-canonical Wnt signaling (β-catenin-independent) pathway involved in a variety of cellular processes [72]. The aberrant activation or inhibition of Wnt5a signaling is emerging as an important event in tumorigenesis, exerting both oncogenic and tumor suppressive effects in cancer. While Wnt5a activation has been shown to inhibit the cell growth, migration, and invasiveness of thyroid and CRC cells [73], increased Wnt5a expression is involved in the aggressiveness of other types of cancers [74]. The tumor suppressor effects of Wnt5a may be achieved by antagonizing canonical Wnt signaling, resulting in the inhibition of cell growth and migration [75]. Another study showed that Wnt5a expression stimulated the directional migration and invasion of colon cancer cells and was correlated with poor prognoses [76]. Further research is needed clarify this apparent discrepancy about the role of Wnt5a signaling in colon cancer.

Moreover, Wnt5a is one endogenous mediator that has been implicated in driving the molecular mechanism of innate tolerance [77]. Mehmeti described the binding of Wnt5a to toll-like receptor 2 (TLR2) under inflammatory conditions. The Wnt5a/TLR2-induced signal promotes the induction of the immunosuppressive cytokine IL-10 in human or pro-inflammatory TNFα in mice [78]. TLRs are overexpressed in some carcinomas, leading to their exploration as prognostic markers and targets for oncological treatment [79]. Interestingly, in our analysis, we observed the downregulation of TLR2, suggesting a link between Wnt5a/TLR2 and EMICORON activity. Moreover, two upregulated genes seem to be promising: peroxiredoxin 4 (PRDX4) and aminoacylase 1 (ACY1).

PRDX4 is a member of peroxiredoxin family involved in a cellular endogenous defense system against ROS, especially in catalyzing peroxide reduction to eliminate excessive cellular H2O2. PRDX4 plays key roles in several cellular functions. It is a multifunctional protein that is involved in protection against oxidative injury, the regulation of cell proliferation, and the modulation of intracellular signaling, and it has also been associated with the pathogenesis of tumors [80].

Moreover, PRDX4 limits inflammation by directly regulating IL-1β generation via the inactivation of caspase-1 activity [81].

PRDX4 has been reported to be over-expressed in CRC and its levels are tightly related to lymph-nodes’ metastases [82]. Conversely, another study identified that the enhanced expression of PRDX4 is associated with the curcumin-enhancing efficacy of the irinotecan-induced apoptosis of colorectal cancer LOVO cells [83].

ACY1 is a cytosolic enzyme that deacylates the α-acylated amino acid from the N-terminal peptide of intracellular proteins [84]. In addition to its function, ACY1 has been studied in different types of human cancer. In small cell lung cancer (SCLC), liver cancer, and renal cell carcinoma, ACY1 expression has been significantly reduced [85], suggesting that ACY1, in these tumors, acts as tumor suppressor. The role of ACY1 in CRC remains unclear. Bing and co-workers [86] observed that the level of ACY1 mRNA and protein is positively associated with the TNM stage, and the inhibition of ACY1 expression leads to the reduction of proliferation and the increase of apoptosis in CRC cells. These results indicate that in CRC, ACY1 has a tumor-promoting function. A further evaluation of ACY1 could lead to its identification as a clinically useful prognostic marker and a potential therapeutic target for CRC.

Of note, the interests towards these genes could be manifold since they could represent both the direct/indirect effectors of EMICORON activity and druggable targets that are useful for potentiating EMICORON activity.

## 7. Conclusions

The current knowledge of the complexity of cancer and the progress of technologies supports the transition from approaches based on simple models to multi-models that are better able to mimic tumors in patients and to guide for advancements in drug development. Moreover, we now have the opportunity to integrate genomic and transcriptomic analyses with advanced experimental models, even if their potential to predict patient clinical outcomes remains to be established. A critical point is represented by the time and cost required—two aspects that make this approach difficult to be pursued in the early stages of drug development when a large number of drugs are screened. However, in our opinion, the use of next-generation strategies could increase the potential success of investigational compounds and should be considered in later stages of preclinical studies, before entering clinical phases.

In conclusion, the integration of omics and advanced mouse models of mCRC with more clinical-like criteria of the evaluation of antitumor efficacy in preclinical studies, could accelerate the anticancer discovery process and provide new weapons against cancer.

## Figures and Tables

**Figure 1 cancers-12-01830-f001:**
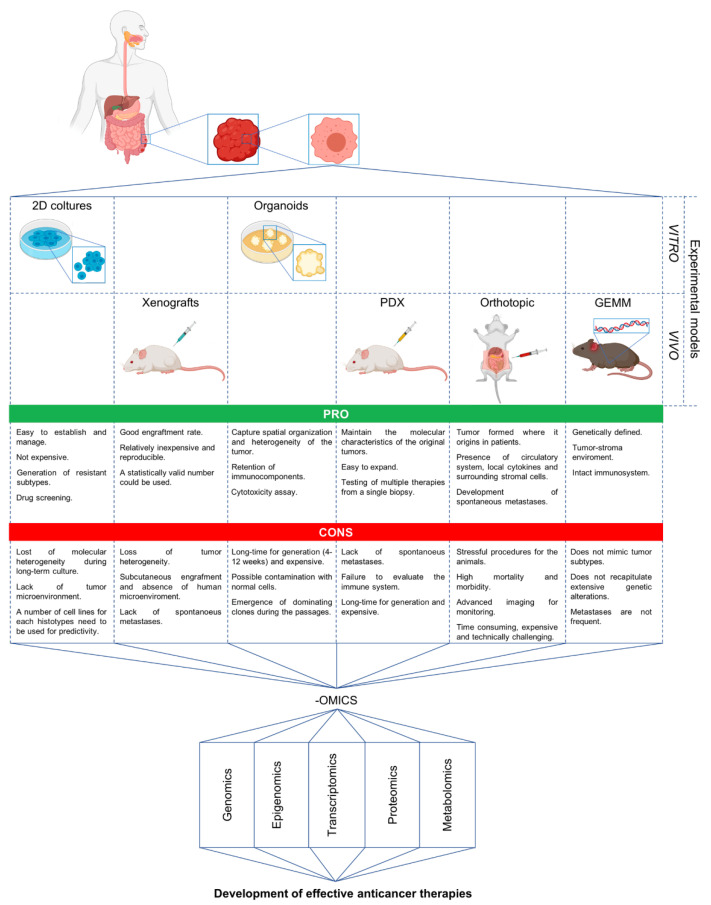
Schematic representation showing the main experimental models and OMICS approaches that can be integrated for the development of effective anticancer drugs. This figure was created with BioRender.com.

**Figure 2 cancers-12-01830-f002:**
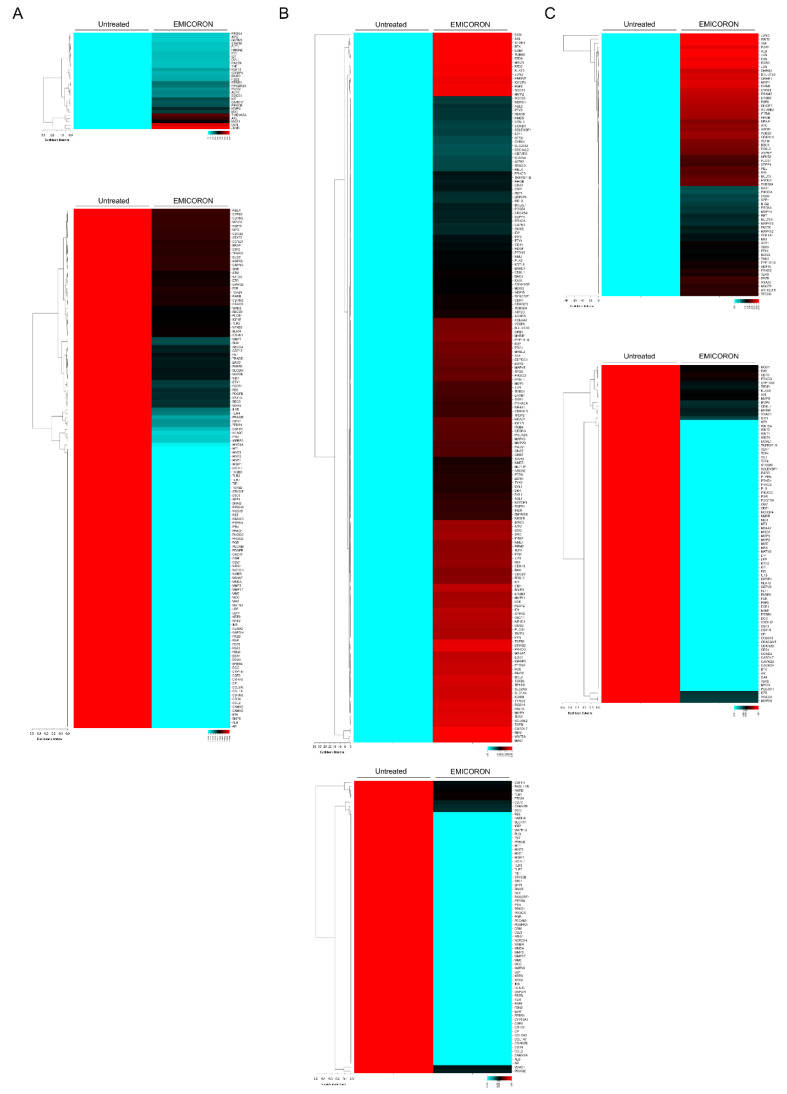
Gene expression analysis in patient-derived-xenografts (PDXs) and in HCT116 cells treated with EMICORON. (**A**) Heatmap representation of cancer-related genes analyzed by TaqMan OpenArray that were up-regulated (upper panel) or down-regulated genes (lower panel) in a responder PDX mouse model treated or untreated with EMICORON. (**B**) Heatmap representation of cancer-related genes analyzed by TaqMan OpenArray that were up-regulated (upper panel) or down-regulated genes (lower panel) in a non-responder PDX mouse model treated or untreated with EMICORON. (**C**) Heatmap representation of cancer-related genes analyzed by TaqMan OpenArray that were up-regulated (upper panel) or down-regulated genes (lower panel) in HCT116 cells that were treated or untreated with EMICORON.

**Figure 3 cancers-12-01830-f003:**
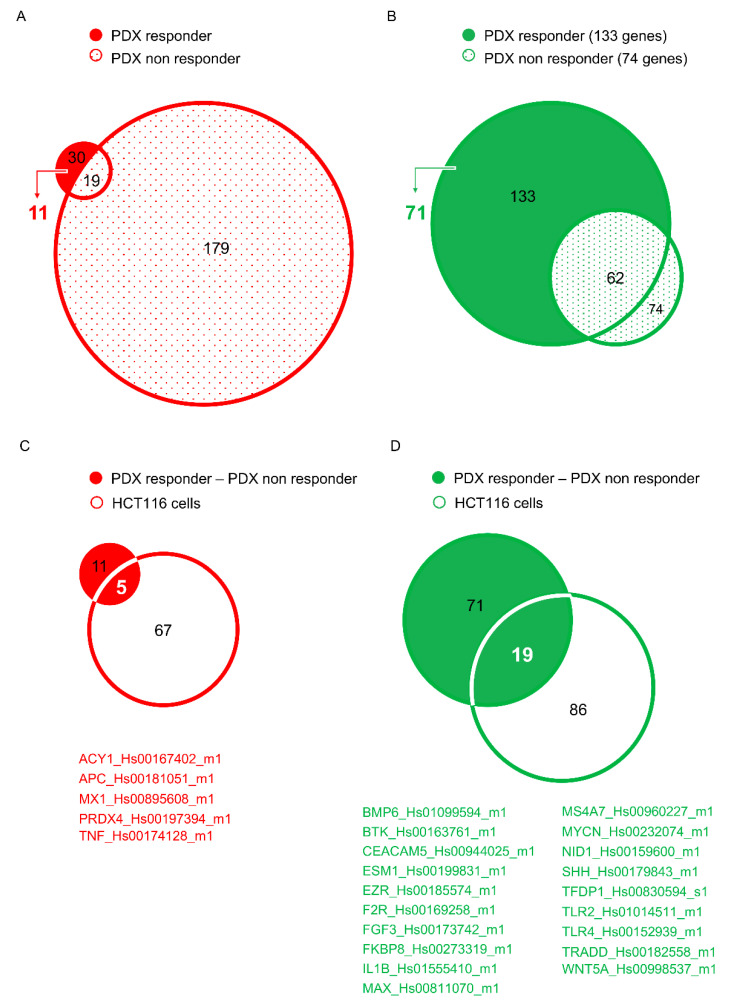
Selection of target genes of EMICORON. (**A**) Venn diagrams showing the overlap between the genes up-regulated in the responder and non-responder PDX models following treatment with EMICORON (given orally at 15 mg/kg/mouse daily for two consecutive weeks). Eleven genes selectively upregulated in the responder mouse were selected as targets of EMICORON. (**B**) Venn diagrams showing the overlap between the genes down-regulated in the responder and non-responder PDXs treated with EMICORON (as above reported). The analysis identified 71 genes targets of EMICORON. (**C**) Venn diagrams showing the overlap between the genes selected in the panel A and the genes up-regulated in the HCT116 cells treated with EMICORON (1 µM for 24 h). The analysis identified 5 common genes that were selected as targets of EMICORON. (**D**) Venn diagrams showing the overlap between the genes selected in the panel B and the genes down-regulated in the HCT116 cells treated with EMICORON (1 µM for 24 h). The analysis identified 19 common genes selected as targets of EMICORON.

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
