# Peer review of "Harnessing Omics Approaches on Advanced Preclinical Models to Discovery Novel Therapeutic Targets for the Treatment of Metastatic Colorectal Cancer"

_cancers, 2020, doi:10.3390/cancers12071830_

Round 1

Reviewer 1 Report

cancers-838398

General Comments

This manuscript describes molecular therapy of colon cancer, based on the accumulating evidence from in vivo and in vitro cancer research. This manuscript is hard to read because of too long sentences. In addition, logic flow of the manuscript seems to be unclear due to lack of appropriate topic sentence and/or discourse markers. Extensive revision would be required to enhance the readability of this review article.

Specific Points

  1. Page 1 Line 2. “Title.” should be deleted.
  2. Page 1 Lines 18-29. Suddenly, another font appeared in the manuscript. Please correct it.
  3. Page 1 Lines 20-24. This sentence is too long to follow (approximately 50 words). Please revise it.
  4. Page 1 Lines 38-44. This sentence is NOT concise. Please revise it.
  5. Page 3 Lines 97-102. This sentence is too long and should be separated into two parts.
  6. Page 3 Lines 117-119. Unclear sentence. Please rewrite it.
  7. Page 3 Lines 140-142. Possibly incomplete sentence. Please rewrite it.
  8. Page 4 Lines 171-175. This sentence is complex. Please revise it
  9. Page 4 Line 173. Please clearly define the abbreviation of “ctDNA” in the text.
  10. Page 4 Lines 181-182. Nevertheless, the reviewer consider that repeated conventional biopsy also enables us to understand the sequential genetic alternation of cancer. Therefore, there is room for amending this unfair sentence.
  11. Pages 5 Line 208. This section 5 contains several topics about different experimental models. Therefore, the reviewer recommends that the authors separate this section into several subsection. In addition, the relationship between this section and the previous sections seems to be somewhat obscure.
  12. Pages 5-6 Lines 216-250. This paragraph is too long. Please revise it.
  13. Page 6 Line 252. Suddenly, another font appeared in the manuscript. Please correct it.
  14. Page 6 Line 262. Please clearly define the abbreviation of “WGS” and “WES” in the text.
  15. Page 7 Line 305. “WGS” has been already defined in the line 262.
  16. Page 7 Line 310. Significance of “BL” and “PD” is unclear. Please expand these abbreviations. And, what does it mean, respectively?
  17. Page 7 Lines 318-319. At a glance, the section title and the first sentence seem to be irrelevant. Therefore, these descriptions lead to readers’ confusion. Please revise them. In addition, the relationship between this section and the previous sections seems to be somewhat obscure.
  18. Figure 1. Although the heatmaps show distinct gene expression provide, each gene is unreadable. Therefore, these images are almost meaningless.
  19. Page 11 Line 439. “-ACY1” seem to be a typo.
  20. Page 11 Line 452. As described previously, the relationship between this section and the previous sections seems to be somewhat obscure. In conclusion, this manuscript consist of scattered various topics and, are NOT organized.
  21. Page 12 Lines 488-494. Because this paragraph and figure 3 are associated with the section 5, the reviewer recommends the authors to move them to near the section 5.
  22. Pages 14-20 Lines 524-854. Suddenly, another font appeared in the manuscript. Please correct it.
  23. Generally, Human genes should be written in italic and all upper cases. In addition, the description of gene mutation is inconsistent, for example, Line 81, 140 and 243.

Author Response

Dear Editor,

we would like to thank the referees for their painstaking work, we did a careful revision of the manuscript in order to address their concerns. We reported below point-by-point answers to referees’ comments

Reviewer #1

We have really appreciated the constructive comments and the suggestions of this referee. In the revised version of the manuscript we have substantially work on the text in order to mainly simplify too long and complex sentences and to clarify the link between the different sections of the work. Text mistakes and inconsistencies have been corrected (the main changes are reported in red within the main text).

Question 1: Page 1 Line 2. “Title.” should be deleted.

Answer 1: “Title” word has been deleted.

Question 2: Page 1 Lines 18-29. Suddenly, another font appeared in the manuscript. Please correct it.

Answer 2: The text is now uniformly formatted.

Question 3: Page 1 Lines 20-24. This sentence is too long to follow (approximately 50 words). Please revise it.

Answer 3: In agreement with the referee's comment the sentence has been simplified. Lines 19-21 of the revised manuscript.

Question 4: Page 1 Lines 38-44. This sentence is NOT concise. Please revise it.

Answer 4: Following the reviewer’s suggestion we reformulated the indicated sentence by dividing it in two parts. Lines 35-40 of the revised manuscript.

Question 5: Page 3 Lines 97-102. This sentence is too long and should be separated into two parts.

Answer 5: In agreement with this reviewer, the sentence has been separated in two parts. Lines 93-97 of the revised manuscript’s version.

Question 6: Page 3 Lines 117-119. Unclear sentence. Please rewrite it.

Answer 6: The sentence has been rewritten to make it clearer. Lines 112-115 of the revised version.

Question 7: Page 3 Lines 140-142. Possibly incomplete sentence. Please rewrite it.

Answer 7: According to the reviewer’s suggestion the sentence has been completed. Lines 136-138 of the revised version.

Question 8: Page 4 Lines 171-175. This sentence is complex. Please revise it

Answer 8: The indicated sentence has been streamlined. Lines 167-171 of the revised manuscript.

Question 9: Page 4 Line 173. Please clearly define the abbreviation of “ctDNA” in the text.

Answer 9: The meaning of “ctDNA” has been clarified in the revised version of the manuscript. Line 168

Question 10: Page 4 Lines 181-182. Nevertheless, the reviewer consider that repeated conventional biopsy also enables us to understand the sequential genetic alternation of cancer. Therefore, there is room for amending this unfair sentence.

Answer 10: We agree that conventional biopsy maintains a crucial role in understanding the sequential genetic alternation of cancer. For this reason, the indicated sentence was removed and a comment was introduced at the end of the paragraph 4. Line 200 of the revised manuscript.

Question 11: Pages 5 Line 208. This section 5 contains several topics about different experimental models. Therefore, the reviewer recommends that the authors separate this section into several subsection. In addition, the relationship between this section and the previous sections seems to be somewhat obscure.

Answer 11: In the revised version of the manuscript we modified the first paragraph of the section 5 in order to better clarify the relationship between this section and the previous ones (Lines 202-208 of the revised manuscript). However, since our work aims at pointing the importance of integrating different preclinical models, we retain more suitable to maintain the original text organization. Moreover, the tight link existing between the different experimental models, would make difficult and very confusing the fragmentation of this section into multiple sub-sections.

Question 12: Pages 5-6 Lines 216-250. This paragraph is too long. Please revise it.

Answer 12: According to the reviewer’s suggestion the paragraph has been condensed. Lines 209-227 of the revised version.

Question 13: Page 6 Line 252. Suddenly, another font appeared in the manuscript. Please correct it.

Answer 13: We are sorry for the mistake; the text is now uniformly formatted.

Question 14: Page 6 Line 262. Please clearly define the abbreviation of “WGS” and “WES” in the text.

Answer 14: We modified the sentence avoiding acronyms. Lines 239-240 of the revised manuscript.

Question 15: Page 7 Line 305. “WGS” has been already defined in the line 262.

Answer 15: Related to answer 14, “whole genome sequencing” has been reported in full. Line 282 of the revised manuscript.

Question 16: Page 7 Line 310. Significance of “BL” and “PD” is unclear. Please expand these abbreviations. And, what does it mean, respectively?

Answer 16: In agreement with the comment of this referee, we decided to shortening and simplify the entire sentence. Lines 285-287 of the revised manuscript.

Question 17: Page 7 Lines 318-319. At a glance, the section title and the first sentence seem to be irrelevant. Therefore, these descriptions lead to readers’ confusion. Please revise them. In addition, the relationship between this section and the previous sections seems to be somewhat obscure.

Answer 17: According to reviewer’s comment, we modified the title and the first sentence of section 6 to make the text less confusing for the readers and to reinforce, at the same time, the link between this section and the previous ones. Lines 304-310 of the revised manuscript.

Question 18: Figure 1. Although the heatmaps show distinct gene expression provide, each gene is unreadable. Therefore, these images are almost meaningless.

Answer 18: We agree with the reviewer that the names of the genes are difficult (if not impossible) to read in the printed version of the manuscript. This is a common problem when large scale analyses (e.g. extensive heatmaps) are presented in a single figure. Conversely, the problem does not exist in the electronic format since high-resolution images can be enlarged without quality loss.

Anyway, the main aim of this figure is that of showing the analysis deriving from the OpenArray while the names of the genes most relevant for our study have been reported in the text.

Question 19: Page 11 Line 439. “-ACY1” seem to be a typo.

Answer 19: This typo has been corrected

Question 20: Page 11 Line 452. As described previously, the relationship between this section and the previous sections seems to be somewhat obscure. In conclusion, this manuscript consist of scattered various topics and, are NOT organized.

Answer 20: We agree with this reviewer that the relationship between this section and the previous ones is weak. Based on this, we deleted the entire section and revised the conclusion section.

Question 21: Page 12 Lines 488-494. Because this paragraph and figure 3 are associated with the section 5, the reviewer recommends the authors to move them to near the section 5.

Answer 21: According to the reviewer’s suggestion, the paragraph and the figure 3 (figure 1 in the revised manuscript) were moved to the end of section 5. Lines 292-302 of the revised manuscript.

Question 22: Pages 14-20 Lines 524-854. Suddenly, another font appeared in the manuscript. Please correct it.

Answer 22: The mistake has been corrected; the text is now uniformly formatted.

Question 23: Generally, Human genes should be written in italic and all upper cases. In addition, the description of gene mutation is inconsistent, for example, Line 81, 140 and 243

Answer 23: In agreement with the indications of the referee, the gene names and mutation have been adjusted within the entire manuscript.

Reviewer 2 Report

Porru et al., reviewed on Harnessing omics approaches on advanced preclinical models to discovery novel therapeutic targets for the treatment of metastatic colorectal cancer. It is a interesting topic in the area of metastatic colon cancer and authors did well in compiling all the available information up to date.

Minor comments:

Spelling mistakes were found at some places. Please check carefully.

Text font is different in abstract.

Author Response

Dear Editor,

we would like to thank the referees for their painstaking work, we did a careful revision of the manuscript in order to address their concerns. We reported below point-by-point answers to referees’ comments

Reviewer #2

We would like to thank this referee for the positive comments and his/her work .

Questions: Spelling mistakes were found at some places. Please check carefully. Text font is different in abstract.

Answer: In agreement with the comments of this reviewer, we carefully revised the text for spelling and formatting mistakes.

Round 2

Reviewer 1 Report

The manuscript has been revised well. The reviewer thinks this manuscript will be acceptable now.